# Follow-Up Differential Descriptions: Language Models Resolve Ambiguities for Image Classification

**Reza Esfandiarpoor & Stephen H. Bach**
Department of Computer Science
Brown University
Providence, RI 02906, USA
{reza_esfandiarpoor,stephen_bach}@brown.edu

## Abstract

A promising approach for improving the performance of vision-language models like CLIP for image classification is to extend the class descriptions (i.e., prompts) with related attributes, e.g., using `brown sparrow` instead of `sparrow`. However, current zero-shot methods select a subset of attributes regardless of commonalities between the target classes, potentially providing no useful information that would have helped to distinguish between them. For instance, they may use color instead of bill shape to distinguish between sparrows and wrens, which are both brown. We propose Follow-up Differential Descriptions (FuDD), a zero-shot approach that tailors the class descriptions to each dataset and leads to additional attributes that better differentiate the target classes. FuDD first identifies the ambiguous classes for each image, and then uses a Large Language Model (LLM) to generate new class descriptions that differentiate between them. The new class descriptions resolve the initial ambiguity and help predict the correct label. In our experiments, FuDD consistently outperforms generic description ensembles and naive LLM-generated descriptions on 12 datasets. We show that differential descriptions are an effective tool to resolve class ambiguities, which otherwise significantly degrade the performance. We also show that high quality natural language class descriptions produced by FuDD result in comparable performance to few-shot adaptation methods. Code: `https://github.com/BatsResearch/fudd`

## 1 Introduction

What is the most distinguishing characteristic of a sparrow? It depends. To distinguish it from what? To distinguish it from a goldfinch, it is the brown color. But, to distinguish it from a wren, it is the conical bill (Fig. 1). Here, we propose a zero-shot approach to adapt the class representations of vision-language models based on other classes in an image classification task. We use natural language descriptions (called *prompts*) to provide visually differentiating information for target classes.

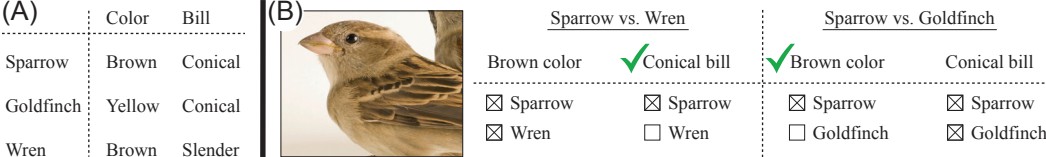

Figure 1: A) Attributes for three different classes. B) Two sample classification tasks involving the wren class. The distinguishing characteristics of each class vary based on other classes. Our approach selects the class descriptions based on other classes in the dataset to provide the information that differentiates the target classes.

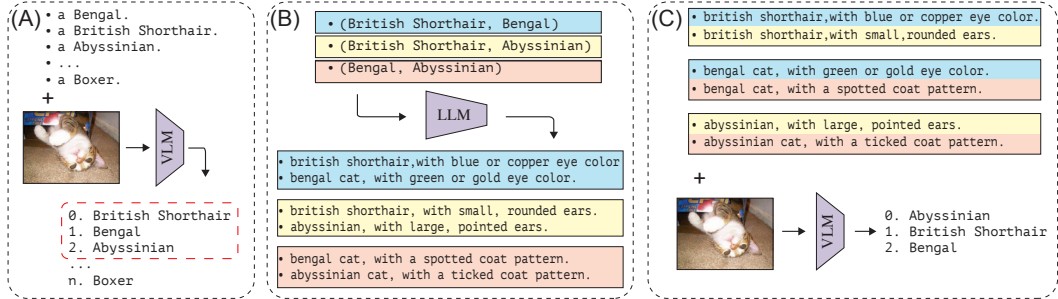

Figure 2: FuDD overview. A) Using the model's initial prediction, we identify the potentially ambiguous classes. B) We use a large language model to generate class descriptions that differentiate the ambiguous classes. C) We use the new differential descriptions in a follow-up classification task to resolve the initial ambiguity and select the correct label.

Large Vision-Language Models (VLMs) use natural language as a source of supervision, which allows us to easily create new classifiers by describing the classes in natural language (e.g., class names) and labeling each image with the closest class. As a result, we can efficiently transfer the learned visual representations to downstream tasks by adapting the class descriptions (i.e., prompts) to the model's pre-training distribution (Chen et al., 2023; Khattak et al., 2023; Menghini et al., 2023; Mirza et al., 2023; Novack et al., 2023; Patashnik et al., 2021; Radford et al., 2021; Zang et al., 2022; Zhang et al., 2023a).

The performance of prompting depends on careful prompt engineering to find class descriptions that provide the most helpful information for each task (Menon & Vondrick, 2023; Novack et al., 2023). Previous works have used class attributes for better zero-shot learning performance (Lampert et al., 2013; Parikh & Grauman, 2011; Romera-Paredes & Torr, 2015; Socher et al., 2013; Xian et al., 2018). Several recent works have adapted this idea for image classification with VLMs and propose to describe the classes by their attributes, such as color and shape (Menon & Vondrick, 2023; Pratt et al., 2022; Yang et al., 2023). Specifically, they prompt a large language model (LLM) with queries like `What does an image of a {class name} look like?`, and use the responses as the new class descriptions. The main idea is to enable the model to use the described attributes in addition to the class names to identify each class. This should lead to better class representations since models can better detect the attributes because of their prevalence during pre-training.

The additional information is helpful if it discriminates the class from all other classes in the target dataset. At least, the provided information should differentiate the class from other classes in the dataset that are frequently mistaken for it. Despite this crucial requirement, current zero-shot approaches generate descriptions solely based on the class itself without considering other classes in the dataset (Menon & Vondrick, 2023; Pratt et al., 2022). As a result, although the descriptions provide additional details about the class, they might not contain any information that differentiates the class from other classes in the dataset. Thus, current methods might generate class descriptions that are not helpful for the target task.

In this paper, we propose Follow-up Differential Descriptions (FuDD[1]), a novel approach that tailors the class descriptions to each dataset and provides additional details that resolve the potential ambiguities in the target task. For each image, we first use a set of basic class descriptions as usual (Radford et al., 2021) and identify a subset of classes that, according to the VLM, are likely to be the true label and thus are considered ambiguous. Then, for each such ambiguous class, we generate a set of descriptions that include the details that differentiate it from other ambiguous classes. We rely on the extended world knowledge of pre-trained large language models to generate such differential descriptions at scale (Brown et al., 2020; Petroni et al., 2019). Finally, we use these differential descriptions in a follow-up classification task to resolve the initial ambiguity and predict the correct label. By customizing the class descriptions based on other classes in the dataset, FuDD aims to provide the most effective information for separating the classes of the target dataset.

---

[1]Pronounced like food.

We evaluate our method on 12 datasets and show that FuDD consistently outperforms naive descriptions for all datasets. FuDD outperforms naive LLM-generated descriptions by 2.41 percentage points on average, with up to 13.95 percentage points improvement for the EuroSAT dataset. In our experiments, we show that not all descriptions resolve ambiguities, and effective class descriptions should provide differentiating information about ambiguous classes. Moreover, differentiating the highly-ambiguous classes is the most important factor, accounting for most of FuDD's performance gains. In addition to GPT-3.5 [2], we experiment with the smaller, publicly available Llama 2 model (Touvron et al., 2023) to study the impact of further fine-tuning, and find that the 7b-parameter model can provide helpful information for everyday concepts. It also benefits from further fine-tuning, especially for rare and abstract concepts in the EuroSAT and DTD datasets, with up to 23.41 percentage points boost in accuracy. Finally, we show that the performance when using high-quality class descriptions from FuDD is comparable to using few-shot methods, achieving performance competitive with 16-shot VLM adaptation methods (Yang et al., 2023; Zhou et al., 2022b) for some datasets. Our results uncover the potential of using natural language to tailor the class representations to each dataset by providing information that differentiates the ambiguous classes. These results motivate future work on creating effective class descriptions for each downstream task.

## 2 RELATED WORK

There is an increasing body of work on adapting VLMs to a wide range of downstream tasks (Gao et al., 2021; Guo et al., 2023; Jia et al., 2022; Novack et al., 2023; Patashnik et al., 2021; Rao et al., 2022; Udandarao et al., 2022; Zeng et al., 2022; Zhang et al., 2021; 2022). Here, we describe the related work and highlight their differences with our method.

**Prompt Tuning**   Prompt tuning is an efficient approach for few-shot adaptation of VLMs to downstream classification tasks. Instead of updating the model parameters, prompt tuning methods add learnable parameters to the input image or text (i.e., prompt) and learn these parameters through gradient descent for each dataset (Huang et al., 2022; Jia et al., 2022; Menghini et al., 2023; Nayak et al., 2022; Zhou et al., 2022a;b). For instance, CoOp adds a set of parameters to the class descriptions to represent the dataset context; and then uses a few labeled samples for training (Zhou et al., 2022b). Although prompt tuning methods achieve good accuracy, they require additional labeled examples, which limits their applications. On the other hand, our method is zero-shot and adapts to each dataset without any additional samples, with competitive performance to prompt-tuning methods in low-shot scenarios.

**VLMs with Other Foundation Models**   One line of work uses the capabilities of other foundation models (Brown et al., 2020; Caron et al., 2021; Ramesh et al., 2021) in combination with VLMs to better adapt to downstream tasks (Chen et al., 2023; Gupta & Kembhavi, 2023; Menon & Vondrick, 2023; Mirza et al., 2023; Pratt et al., 2022; Surís et al., 2023; Zeng et al., 2022; Zhang et al., 2023b). For example, one can use the extended world knowledge of large language models (LLMs) in combination with VLMs to solve more complex visual tasks. Our approach is closely related to this line of work; we discuss the differences further in the next paragraph. Several other methods use text-to-image generation models (Rombach et al., 2022) on top of LLMs (Brown et al., 2020) to further improve the performance (Udandarao et al., 2022; Zhang et al., 2023a). For instance, SuS-X first uses an LLM to generate class descriptions and then uses a text-to-image generation model to generate synthetic images for each class based on these descriptions (Udandarao et al., 2022). Our experiments show that despite using no images, FuDD's performance is comparable to SuS-X for most datasets while avoiding the complexities of text-to-image generation models.

**Adaptation Through Description**   A specific approach for improving class representations without additional samples is to provide more informative class descriptions (Menon & Vondrick, 2023; Novack et al., 2023; Pratt et al., 2022; Roth et al., 2023). For example, WaffleCLIP adds high-level category names to class descriptions to avoid ambiguities caused by class names with multiple meanings (Roth et al., 2023). Another approach is to describe the classes with their attributes so the model can rely on attributes in addition to class names to identify images of each class (Menon & Vondrick, 2023; Pratt et al., 2022). For example, Menon & Vondrick (2023) propose to generate such class descriptions by querying an LLM about the most important attributes of each class. However, the generated descriptions might provide no useful information for separating the class from

---

[2] https://openai.com/blog/openai-api

other classes in the dataset. For example, the attribute color is not useful for separating sparrows and wrens since both are brown (Fig. 1). To address this issue, LaBo uses additional labeled examples to learn the importance of each attribute (Yang et al., 2023). Then, it selects a set of attributes that are the most discriminative for each class. Unlike LaBo, FuDD generates class descriptions that effectively separate the target classes in the first place, eliminating the need for further optimization. Despite using no labeled data, FuDD's performance is comparable to few-shot LaBo for most datasets.

## 3 Follow-up Differential Descriptions (FuDD)

Here, we describe the components of our proposed method, FuDD. In Section 3.1, we explain how VLMs are used for image classification. In Section 3.2, we use the model's initial predictions to identify potentially misrepresented classes that could lead to misclassifications, i.e., are ambiguous (Fig. 2a). In Section 3.3, we use large language models to generate class descriptions that explain the visually differentiating information for the ambiguous classes (Fig. 2b). In Section 3.4, we use these differential descriptions in a follow-up classification task to resolve the initial ambiguity (Fig. 2c).

### 3.1 Background

Following previous work (Radford et al., 2021), given a set of classes, $C$, and a set of descriptions, $D_c$, for each class, we calculate the class embeddings as:

$$h_c = \frac{1}{|D_c|} \sum_{d \in D_c} \phi_T(d) \,,$$

where $\phi_T$ is the VLM text encoder, and $h_c$ is the embedding vector for class $c$. Since VLMs are trained to minimize the distance between related image-text pairs, we select the closest class to image embedding $\phi_I(x)$ as the label for image $x$, where $\phi_I$ is the VLM image encoder.

### 3.2 Detecting Ambiguous Classes

Enumerating the differences between all class pairs is prohibitive for large datasets with thousands of classes. Instead, we focus on a small subset of potentially ambiguous classes that can lead to misclassifications. For example, in Fig. 2a, the model is confident that boxer (a dog breed) is not the label. However, any of the three most similar classes (british shorthair, bengal, and abyssinian cat) is likely to be the true label. Therefore, differentiating visual information for these classes is sufficient for selecting the correct label. For an image $x$, we define the set of ambiguous classes $C_A$, as the $k$ most similar classes:

$$C_A = \underset{\{c_1, \ldots, c_k\} \subseteq C}{\arg\max} \sum_{c_i} \cos(\phi_I(x), h_{c_i}) \,,$$

where $\phi_I$ is the VLM image encoder, and $\cos$ is the cosine similarity operator.

### 3.3 Differential Descriptions

To help the model distinguish between ambiguous classes, we generate a set of class descriptions that explain their visual differences. We take advantage of the extended world knowledge of LLMs to generate such descriptions at scale. Despite being uni-modal, LLMs acquire knowledge of the visual world through massive pre-training datasets (Jiang et al., 2020; Petroni et al., 2019). For example, an LLM can learn how a sparrow looks by reading the related Wikipedia page.

For each pair of ambiguous classes, we condition the LLM to select the visually differentiating attributes and describe them for both classes. We use the in-context learning capabilities of LLMs (Brown et al., 2020) to guide the model to focus on visual characteristics by providing two fixed examples as part of the prompt. Similarly, we guide the LLM to generate descriptions that resemble photo captions, which is shown to better adapt to VLMs' pre-training distributions (Radford et al., 2021) (refer to appendix for more details). We use the following prompt template:

Table 1: Accuracy of FuDD in comparison with baselines. B/32 and L/14[*] represent the ViT-B/32 and ViT-L/14@336px vision backbones. $\Delta$Naive$(k)$ is the improvement of FuDD with $k$ ambiguous classes over the Naive LLM-generated descriptions proposed by Menon & Vondrick (2023).

| Description | Cub | | DTD | | EuroSAT | | FGVCAircraft | | Flowers102 | | Food101 | |
|---|---|---|---|---|---|---|---|---|---|---|---|---|
| | B/32 | L/14[*] | B/32 | L/14[*] | B/32 | L/14[*] | B/32 | L/14[*] | B/32 | L/14[*] | B/32 | L/14[*] |
| Single Template | 51.21 | 63.48 | 43.14 | 54.04 | 40.87 | 56.82 | 20.88 | 37.08 | 63.80 | 75.12 | 82.63 | 93.49 |
| Template Set | 51.52 | 64.07 | 42.71 | 55.32 | 46.76 | 54.27 | 21.15 | 38.31 | 63.44 | 74.14 | 83.16 | 93.77 |
| Naive LLM | 52.92 | 65.15 | 45.90 | 55.37 | 44.18 | 46.69 | 21.09 | 38.79 | 66.12 | 75.98 | 84.02 | 94.26 |
| FuDD ($k$=10) | 53.97 | 65.90 | 45.43 | 57.66 | 45.18 | 60.64 | 21.87 | 38.82 | 67.80 | 78.76 | 84.05 | 94.05 |
| FuDD ($k$=$|C|$) | 54.30 | 66.03 | 44.84 | 57.23 | 45.18 | 60.64 | 22.32 | 39.63 | 67.62 | 79.67 | 84.36 | 94.27 |
| $\Delta$ Naive ($k$=10) | ↑1.05 | ↑0.75 | ↓-0.47 | ↑2.29 | ↑1.00 | ↑13.95 | ↑0.78 | ↑0.03 | ↑1.68 | ↑2.78 | ↑0.03 | ↓-0.21 |
| $\Delta$ Naive ($k$=$|C|$) | ↑1.38 | ↑0.88 | ↓-1.06 | ↑1.86 | ↑1.00 | ↑13.95 | ↑1.23 | ↑0.84 | ↑1.50 | ↑3.69 | ↑0.34 | ↑0.01 |

| | ImageNet | | ImageNet V2 | | Oxford Pets | | Places365 | | Stanford Cars | | Stanford Dogs | |
|---|---|---|---|---|---|---|---|---|---|---|---|---|
| | B/32 | L/14[*] | B/32 | L/14[*] | B/32 | L/14[*] | B/32 | L/14[*] | B/32 | L/14[*] | B/32 | L/14[*] |
| Single Template | 62.04 | 74.85 | 54.77 | 68.79 | 84.98 | 92.86 | 39.10 | 40.70 | 60.37 | 78.06 | 58.01 | 73.61 |
| Template Set | 63.37 | 76.54 | 55.91 | 70.85 | 84.55 | 92.70 | 40.91 | 42.54 | 60.38 | 79.12 | 57.79 | 74.01 |
| Naive LLM | 63.52 | 76.37 | 55.96 | 70.47 | 83.76 | 93.08 | 40.58 | 41.43 | 59.63 | 77.90 | 57.86 | 74.02 |
| FuDD ($k$=10) | 64.05 | 76.70 | 56.62 | 70.60 | 86.92 | 93.40 | 42.12 | 43.95 | 60.86 | 78.25 | 60.03 | 75.99 |
| FuDD ($k$=$|C|$) | 64.19 | 77.00 | 56.75 | 71.05 | 89.34 | 93.51 | 42.17 | 44.09 | 61.46 | 78.96 | 60.28 | 76.34 |
| $\Delta$ Naive ($k$=10) | ↑0.53 | ↑0.33 | ↑0.66 | ↑0.13 | ↑3.16 | ↑0.32 | ↑1.54 | ↑2.52 | ↑1.23 | ↑0.35 | ↑2.17 | ↑1.97 |
| $\Delta$ Naive ($k$=$|C|$) | ↑0.67 | ↑0.63 | ↑0.79 | ↑0.58 | ↑5.58 | ↑0.43 | ↑1.59 | ↑2.66 | ↑1.83 | ↑1.06 | ↑2.42 | ↑2.32 |

```
For the following objects, generate captions that represent the
    distinguishing visual differences between the photos of the two
    objects. Generate as many captions as you can.
Object 1: {class name 1}
Object 2: {class name 2}
```

Following the provided samples, the model generates several responses similar to:

```
Visual characteristic: Bill color
Caption 1: A photo of a black-footed albatross, with a yellow bill.
Caption 2: A photo of a laysan albatross, with a pink bill.
```

Given a pair of classes $c_1$ and $c_2$, we define the pairwise differential descriptions for class $c_1$, $D_{c_1}^{c_2}$, as all the values for `Caption 1` in the LLM response, and similarly define $D_{c_2}^{c_1}$. As a result, $D_{c_1}^{c_2}$ contains all the descriptions that visually distinguish $c_1$ from $c_2$. For each ambiguous class $c$, we combine all its pairwise descriptions to obtain the set of differential descriptions $D_c'$

$$D_c' = \bigcup_{c_i \in C_A \setminus \{c\}} D_c^{c_i} \, .$$

The new set of differential descriptions, $D_c'$, contains all the information necessary for separating class $c$ from other ambiguous classes.

### 3.4 FOLLOW-UP CLASSIFICATION

Since this visually differentiating information resolves the initial ambiguity, after the first round of classification based on the original class descriptions, we create a follow-up classification task with only the ambiguous classes, $C_A$, and the new differential descriptions, $D_c'$. Finally, we follow the steps in Section 3.1 to predict the label.

## 4 EXPERIMENTS

In this section, we show the effectiveness of FuDD through extensive experiments. We show that FuDD outperforms both generic and naive LLM-generated description ensembles. We design further

Table 2: Accuracy of differential and non-differential descriptions for ambiguous classes. B/32 and L/14* represent the ViT-B/32 and ViT-L/14@336px vision backbones. $\Delta$ is the improvement of differential over non-differential descriptions.

| Descriptor | CUB | | DTD | | FGVCAircraft | | Flowers102 | | Food101 | |
|---|---|---|---|---|---|---|---|---|---|---|
| | B/32 | L/14* | B/32 | L/14* | B/32 | L/14* | B/32 | L/14* | B/32 | L/14* |
| Differential | 53.62 | 65.79 | 45.37 | 56.91 | 22.17 | 39.06 | 67.62 | 79.54 | 84.17 | 94.34 |
| Non-Differential | 52.28 | 64.38 | 42.82 | 56.44 | 22.14 | 36.90 | 65.73 | 77.74 | 83.92 | 94.02 |
| $\Delta$ | ↑1.35 | ↑1.42 | ↑2.55 | ↑0.47 | ↑0.03 | ↑2.16 | ↑1.89 | ↑1.81 | ↑0.25 | ↑0.32 |

| Descriptor | Oxford Pets | | Places365 | | Stanford Cars | | Stanford Dogs | |
|---|---|---|---|---|---|---|---|---|
| | B/32 | L/14* | B/32 | L/14* | B/32 | L/14* | B/32 | L/14* |
| Differential | 87.24 | 93.68 | 42.45 | 44.26 | 60.90 | 79.39 | 60.31 | 75.96 |
| Non-Differential | 86.24 | 93.62 | 41.73 | 43.98 | 60.74 | 78.55 | 59.30 | 75.41 |
| $\Delta$ | ↑1.01 | ↑0.06 | ↑0.73 | ↑0.28 | ↑0.16 | ↑0.85 | ↑1.01 | ↑0.55 |

analytical experiments to show that not all semantic information resolves class ambiguities, and effective class descriptions should provide information that differentiates the ambiguous classes. Additionally, we find that describing the differences between highly ambiguous classes is the most important, accounting for most of FuDD's performance gains.

**Datasets.** We evaluate our method on 12 image recognition datasets. We use the CUB200-2011 (Wah et al., 2011) (fine-grained bird species), Describable Textures Dataset (DTD) (Cimpoi et al., 2014) (texture classification), EuroSAT (Helber et al., 2019) (satellite image classification), FGVCAircraft (Maji et al., 2013) (aircraft model classification), Flowers102 (Nilsback & Zisserman, 2008), Food101 (Bossard et al., 2014), ImageNet (Deng et al., 2009), ImageNetV2 (Kornblith et al., 2019), Oxford IIIT Pets (Parkhi et al., 2012), Places365 (Zhou et al., 2017), Stanford Cars (Krause et al., 2013), and Stanford Dogs (Khosla et al., 2011) datasets.

**Setup.** We use an instruction-tuned GPT-3 model (Brown et al., 2020; Ouyang et al., 2022), `gpt-3.5-turbo-0301`, which is available through OpenAI API [3] as our LLM, and CLIP (Radford et al., 2021) as our VLM (refer to appendix for results with other VLMs). **ImageNet Descriptions:** Because of the large number of classes in ImageNet, to accommodate the API limitations, we cache the pairwise descriptions only for ambiguous classes detected by the ViT-B/32 backbone. In all experiments, we limit the available differential descriptions to these cached values (refer to appendix for details).

**Baselines.** We use three baselines. **Single Template:** to adapt to CLIP's pre-training distribution, we adopt `A photo of a {class name}.` as the class description (Radford et al., 2021). **Template Set:** we use the 80 generic templates proposed by Radford et al. (2021) to study the benefits of FuDD's better semantic information beyond simple prompt ensembling. **Naive LLM:** we follow Menon & Vondrick (2023) to create naive LLM-generated descriptions with the same LLM as ours, which uses a prompt like `What are useful features for distinguishing a {class name} in a photo?` with a few in-context examples.

## 4.1 RESULTS

The benefits of using FuDD's semantic information exceed simple description ensembling (Table 1). When descriptions are provided for all classes ($k=|C|$), FuDD outperforms the generic template set on 11 out of 12 datasets with ViT-B/32 (base) and ViT-L/14@336px (large) backbones. Moreover, FuDD is more effective than naive LLM-generated descriptions at resolving class ambiguities. On average, FuDD outperforms naive LLM-generated descriptions by 1.44% and 2.41% with base and large vision backbones, respectively, with up to 13.95% improvements on EuroSAT. Notably, when using the base vision backbone, naive LLM-generated descriptions perform worse than the generic template set on the FGVCAircraft, Oxford Pets, Places365, and Stanford Cars datasets. On the other

---

[3] https://openai.com/blog/openai-api

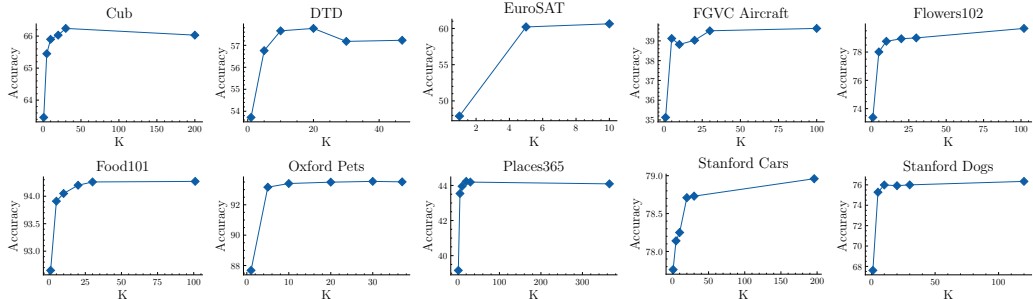

Figure 3: Impact of differential descriptions for $k$ most ambiguous classes with ViT-L/14@336px. $k=1$ is accuracy with a single template. Providing differentiating details for the most ambiguous classes accounts for most of FuDD's gains, with diminishing gains for less ambiguous classes.

hand, FuDD improves accuracy compared to the generic template set by providing differentiating information that resolves ambiguities. Importantly, we observe similar improvements with $k=10$, where FuDD only describes the differences between the 10 most ambiguous classes. This emphasizes the significant impact of class ambiguities on accuracy, which allows computational efficiency, that we will discuss in more detail later.

**Importance of Differential Descriptions** To study the importance of differentiating information, we compare differential descriptions with non-differential descriptions, which describe characteristics that do not separate the ambiguous classes. We select non-differential descriptions from attributes not used by differential descriptions. To control for the number of descriptions, we augment the descriptions with approx. 80 prefixes like image and snapshot, with minimal impact on semantic information (refer to appendix for details). As shown in Table 2, non-differential descriptions perform worse than differential descriptions. Non-differential descriptions lead to lower accuracy by at least 1% for six datasets, with up to 2.16% and 2.55% for FGVCAircraft and DTD. These results confirm that not all semantic information resolves class ambiguities, and effective class descriptions like FuDD should provide the necessary information to differentiate the ambiguous classes.

**Role of Class Ambiguities** Since FuDD mainly focuses on ambiguous classes, here, we examine the importance of resolving class ambiguities. Figure 3 plots the accuracy of FuDD for the $k$ most ambiguous classes for different values of $k$, which corresponds to varying levels of ambiguity (Section 3.2). $k = 1$ is accuracy with a single template. We find that describing the differences between the five most ambiguous classes accounts for most of FuDD's performance gains, with diminishing benefits for less ambiguous classes. We can thus get most of the benefit while avoiding high computational costs, especially in the case of diverse datasets and open-set problems.

## 5  PUBLICLY AVAILABLE LANGUAGE MODELS

The inaccessibility of proprietary LLMs like GPT-3.5 hinders further research into fine-tuning LLMs for visual classification. Here, we use publicly available LLMs to generate the descriptions and study the impact of fine-tuning. Specifically, we fine-tune the 7b-parameter Llama 2 model (Touvron et al., 2023) on the descriptions generated by GPT-3.5 for the ImageNet dataset. We find that even the original Llama 2 model provides useful semantic information for visual classification. Moreover, fine-tuning improves Llama 2 performance, especially for rare concepts like satellite images, achieving comparable performance to GPT-3.5.

Table 3: The percentage of descriptions in a random sample that are correct and visually differentiating for EuroSAT.

| Model | Correct | Useful |
|---|---|---|
| Llama 2 | 82.26 | 19.35 |
| Llama 2 FT | 90.52 | 48.28 |

As reported in Table 4, the original Llama 2 provides helpful semantic information for everyday objects like flowers, outperforming the generic template set. However, it struggles to describe the visual differences for datasets with rare objects like EuroSAT or abstract concepts like DTD. Although the fine-tuned model is not trained on test datasets, it learns the structure of the task and provides differ-

Table 4: FuDD's accuracy with Llama 2 generated descriptions before and after fine-tuning. B/32 and L/14[*] represent the ViT-B/32 and ViT-L/14@336px vision backbones.

| Model | Cub | | DTD | | EuroSAT | | FGVCAircraft | |
|---|---|---|---|---|---|---|---|---|
| | B/32 | L/14[*] | B/32 | L/14[*] | B/32 | L/14[*] | B/32 | L/14[*] |
| Llama 2 ($k$=10) | 53.14 | 64.12 | 41.91 | 54.47 | 27.71 | 37.78 | 21.03 | 38.64 |
| Llama 2 ($k$=$|C|$) | 53.33 | 64.65 | 41.91 | 54.89 | 27.71 | 37.78 | 20.76 | 37.86 |
| Llama 2 FT ($k$=10) | 53.45 | 64.81 | 42.66 | 56.54 | 39.14 | 61.19 | 22.14 | 38.31 |
| Llama 2 FT ($k$=$|C|$) | 53.37 | 64.43 | 43.14 | 56.17 | 39.14 | 61.19 | 22.44 | 39.57 |
| | Flowers102 | | Food101 | | Oxford Pets | | Stanford Cars | |
| | B/32 | L/14[*] | B/32 | L/14[*] | B/32 | L/14[*] | B/32 | L/14[*] |
| Llama 2 ($k$=10) | 66.03 | 77.88 | 83.54 | 94.00 | 87.19 | 93.19 | 60.27 | 77.53 |
| Llama 2 ($k$=$|C|$) | 66.16 | 78.00 | 84.08 | 94.15 | 89.34 | 93.24 | 60.48 | 77.95 |
| Llama 2 FT ($k$=10) | 65.98 | 77.67 | 84.32 | 94.28 | 86.05 | 92.67 | 60.07 | 78.20 |
| Llama 2 FT ($k$=$|C|$) | 66.55 | 77.51 | 84.52 | 94.25 | 87.49 | 92.31 | 61.05 | 79.06 |

entiating visual information for these datasets. Using the ViT-L/14@336px backbone, fine-tuning improves the accuracy by 3.25% on average, with up to 23.41% for EuroSAT.

To better understand the impact of fine-tuning, we manually evaluate a random subset of pairwise differential descriptions before and after fine-tuning. Through visual inspection, for each pair, we check 1) if each description is correct and 2) if the pair helps differentiate the images of the two classes. Although fine-tuning helps with both measures, it significantly improves the usefulness of the descriptions: as shown in Table 3, after fine-tuning, 48% of pairwise descriptions help differentiate the two classes, compared to only 19% before fine-tuning. As illustrated in Fig. 5, unlike the original model, the fine-tuned model describes attributes that are more diverse and focused on low-level visual features rather than higher-level semantic concepts. For a more robust analysis, Fig. 4 plots the top-5 most common attributes before and after fine-tuning for EuroSAT. Similarly, the original model describes a limited set of attributes with mostly high-level semantic information, while the fine-tuned model generates a diverse set of visually differentiating attributes based on the input classes.

## 6 FuDD vs. Other Auxiliary Information

To put the importance of differential descriptions in perspective, we compare FuDD against other approaches for VLM adaptation. We show that FuDD provides more helpful information through differential descriptions compared to simple heuristics like using high-level category names. We find that FuDD can better use the potential of natural language descriptions and achieve comparable performance to other methods that use text-to-image generation models or labeled samples in low-shot settings.

**Few-Shot Description Selection** We compare FuDD against LaBo, an alternative method that uses few-shot learning to select a subset of naive LLM-generated class descriptions that are more discriminative (Yang et al., 2023). As reported in Table 5, although FuDD is zero-shot and uses no

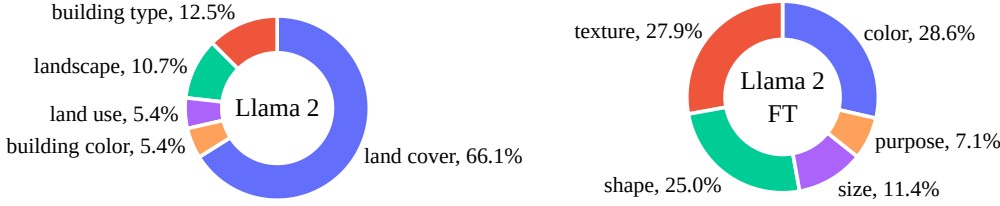

Figure 4: Top-5 most common attributes described by Llama 2 before and after fine-tuning. The fine-tuned model describes a more diverse and visually differentiating set of attributes.

Table 5: Accuracy of FuDD compared to LaBo (Yang et al., 2023), which uses few-shot learning to select the most effective LLM-generated descriptions for each class with ViT-L/14 backbone. #S is the number of labeled samples.

| Method | Cub | | DTD | | Aircraft | | Flowers102 | | Food101 | | ImageNet | |
|---|---|---|---|---|---|---|---|---|---|---|---|---|
| | Acc. | #S | Acc. | #S | Acc. | #S | Acc. | #S | Acc. | #S | Acc. | #S |
| FuDD($k$=10) | 64.26 | 0 | 56.76 | 0 | 37.38 | 0 | 78.61 | 0 | 93.28 | 0 | 75.67 | 0 |
| FuDD($k$=$|C|$) | 64.14 | 0 | 57.07 | 0 | 37.89 | 0 | 79.48 | 0 | 93.40 | 0 | 75.99 | 0 |
| LaBo | 54.19 | 1 | 55.26 | 2 | 37.71 | 2 | 82.05 | 1 | 92.45 | Full | 72.60 | 16 |

labeled images, it performs better than 16-shot and full-shot (training on all samples) LaBo on ImageNet and Food101, respectively. For the other four datasets, FuDD's performance is comparable to LaBo in low-shot scenarios. Unlike LaBo, FuDD encourages the descriptions to be discriminative as part of the generation process, eliminating the need for further optimization.

**High Level Concepts**   WaffleCLIP (Roth et al., 2023) uses high-level category names to address class ambiguities by specifying the dataset context. As shown in Table 6 in appendix, FuDD performs better than WaffleCLIP for seven of the eight datasets. Although high-level category information is helpful, the additional details provided by FuDD are necessary to resolve more complex class ambiguities beyond what is caused by similar class names.

**Additional Images**   In Table 6 in appendix, we compare FuDD with CoOp (Zhou et al., 2022b), which uses additional labeled images to learn a set of parameters as part of class descriptions. Without using any images, FuDD performs better than 16-shot CoOp on Food101 and Oxford Pets datasets and better than 4-shot CoOp on the ImageNet dataset. On the other five datasets, FuDD's performance is comparable to CoOp in low-shot settings. We also compare FuDD with SuS-X (Udandarao et al., 2022), which avoids the additional labeled images by using a pre-trained LLM (Brown et al., 2020) and a text-to-image generation model (Rombach et al., 2022) to generate additional images for each class. As reported in Table 6 in appendix, despite using no images, FuDD achieves a performance comparable to SuS-X by only relying on the LLM-generated descriptions. FuDD uses the potential of natural language more effectively through differential descriptions and achieves comparable performance without additional labeled data or complexities of using text-to-image generation models.

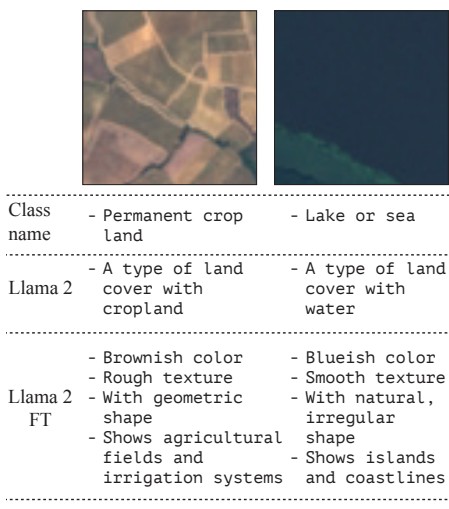

| Class name | - Permanent crop land | - Lake or sea |
|---|---|---|
| Llama 2 | - A type of land cover with cropland | - A type of land cover with water |
| Llama 2 FT | - Brownish color
- Rough texture
- With geometric shape
- Shows agricultural fields and irrigation systems | - Blueish color
- Smooth texture
- With natural, irregular shape
- Shows islands and coastlines |

Figure 5: Descriptions generated by Llama 2 before and after fine-tuning.

## 7 CONCLUSION

In this work, we introduce FuDD, a novel zero-shot approach that uses natural language to provide vision-language models with differentiating information about classes in downstream image recognition tasks. FuDD identifies a potentially ambiguous subset of classes and uses a large language model to generate visually differentiating descriptions that resolve the ambiguity. We show that not all information helps resolve class ambiguities, and effective descriptions should provide discriminative information about the ambiguous classes. Well-designed class descriptions, such as the ones produced by FuDD, can achieve comparable performance to few-shot prompt tuning methods in low-shot settings. Our results uncover the potential of natural language for tailoring the class representations to each dataset by providing differentiating information about ambiguous classes. These results motivate future work on creating effective natural language class descriptions for each downstream task.

ACKNOWLEDGEMENTS

This material is based on research sponsored by Defense Advanced Research Projects Agency (DARPA) and Air Force Research Laboratory (AFRL) under agreement number FA8750-19-2-1006. The U.S. Government is authorized to reproduce and distribute reprints for Governmental purposes notwithstanding any copyright notation thereon. The views and conclusions contained herein are those of the authors and should not be interpreted as necessarily representing the official policies or endorsements, either expressed or implied, of Defense Advanced Research Projects Agency (DARPA) and Air Force Research Laboratory (AFRL) or the U.S. Government. We gratefully acknowledge support from Google and Cisco. Disclosure: Stephen Bach is an advisor to Snorkel AI, a company that provides software and services for data-centric artificial intelligence.

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

Table 6: Accuracy of FuDD compared to other approaches that use high-level category names, WaffleCLIP (Roth et al., 2023), labeled samples, CoOp (Zhou et al., 2022b), and text-to-image generation models, SuS-X (Udandarao et al., 2022), with ResNet-50 He et al. (2016) backbone. #S is the number of labeled samples. *SuS-X-SD uses synthetically generated images.

| Method | Cub | | DTD | | EuroSAT | | Aircraft | | Flowers102 | | Food101 | |
|---|---|---|---|---|---|---|---|---|---|---|---|---|
| | Acc. | #S | Acc. | #S | Acc. | #S | Acc. | #S | Acc. | #S | Acc. | #S |
| FuDD($k$=10) | 49.45 | 0 | 43.51 | 0 | 39.42 | 0 | 19.77 | 0 | 67.39 | 0 | 80.65 | 0 |
| FuDD($k$=$|C|$) | 49.26 | 0 | 43.51 | 0 | 39.42 | 0 | 19.92 | 0 | 68.76 | 0 | 80.95 | 0 |
| WaffleCLIP | 48.34 | 0 | 39.25 | 0 | 35.08 | 0 | - | - | - | - | 81.38 | 0 |
| CoOp | - | - | 44.39 | 1 | 50.63 | 1 | 18.68 | 2 | 68.12 | 1 | 74.67 | 16 |
| SuS-X-SD* | 49.10 | 2 | 51.00 | 4 | 47.69 | 15 | 19.92 | 79 | 67.32 | 31 | 77.02 | 34 |

| | ImageNet | | ImageNetV2 | | Oxford Pets | | Places365 | | Stanford Cars | |
|---|---|---|---|---|---|---|---|---|---|---|
| | Acc. | #S | Acc. | #S | Acc. | #S | Acc. | #S | Acc. | #S |
| FuDD($k$=10) | 60.69 | 0 | 53.19 | 0 | 86.86 | 0 | 40.64 | 0 | 56.62 | 0 |
| FuDD($k$=$|C|$) | 60.78 | 0 | 53.60 | 0 | 87.52 | 0 | 40.69 | 0 | 56.77 | 0 |
| WaffleCLIP | 60.12 | 0 | 52.89 | 0 | 85.80 | 0 | 39.03 | 0 | - | - |
| CoOp | 59.99 | 4 | - | - | 87.01 | 16 | - | - | 55.59 | 1 |
| SuS-X-SD* | 61.65 | 36 | - | - | 85.09 | 71 | - | - | 57.14 | 5 |

## A DIFFERENT VISION ENCODERS

In general, the benefits of FuDD over the generic template set are more significant for smaller models (Fig. 6a). However, larger vision encoders can better take advantage of the nuanced information provided by FuDD beyond naive LLM-generated descriptions (Fig. 6b). We believe as image-text representations improve, VLMs can better take advantage of available semantic information, making natural language class descriptions even more important for vision tasks in the future.

## B LLM PROMPTING DETAILS

We use `gpt-3.5-turbo-0301` to generate the differential descriptions. `gpt-3.5-turbo-0301` is a GPT-3 model that is fine-tuned to follow instructions (Brown et al., 2020; Ouyang et al., 2022). In this form, the model is given a sequence of user and assistant messages and is expected to generate the next assistant message. In our experiments, we encode two fixed sample outputs as assistant messages and ask the model to generate similar output for a given pair of classes. Specifically, we use the template messages in Table 9, and replace `class name 1` and `class name 2` with the desired classes to get the corresponding pairwise differential descriptions.

**ImageNet Descriptions.** As mentioned in Section 4, because of the large number of classes in the ImageNet dataset, it is not possible to generate the differential descriptions for all class pairs. Since

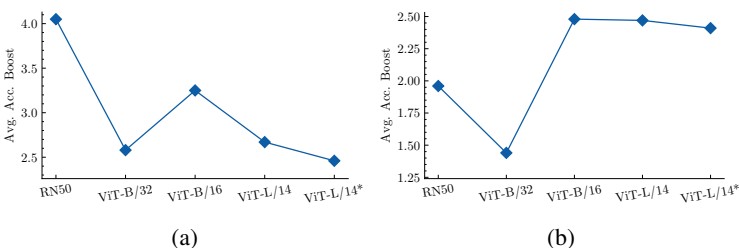

(a)                                           (b)

Figure 6: FuDD's average accuracy boost for different vision backbones compared to a) generic template set and b) naive LLM-generated descriptions

Table 7: Total API costs for each dataset in Dollars

| Dataset | Cost |
|---|---|
| Cub | 15.47 |
| DTD | 0.84 |
| EuroSAT | 0.03 |
| FGVCAircraft | 3.85 |
| Flowers | 4.00 |
| Food101 | 3.93 |
| ImageNet | 60.24 |
| Pets | 0.52 |
| Places365 | 51.63 |
| Stanford Cars | 14.85 |
| Stanford Dogs | 5.55 |

differential descriptions are the most effective for ambiguous classes, we generate and cache all the pairwise descriptions for ambiguous classes and limit the available pairwise differential descriptions to this set. Following Section 3.2, we use ViT-B/32 vision backbone with $k = 5$ to detect the most ambiguous classes. For all pairs in each set of ambiguous classes, we generate and cache the corresponding pairwise differential descriptions, $D_c^{c_i}$, as explained in Section 3.3. In all our experiments, we use the cached pairwise differential descriptions, $D_c^{c_i}$. If the cache does not exist for a class pair, we use a single template description instead, i.e., $D_c^{c_i} = \{$`A photo of a class name.`$\}$.

## C  DIFFERENTIAL VS. NON-DIFFERENTIAL DETAILS

For the non-differential experiments in Table 2, we select a set of descriptions that describe a set of details for each class that does not differentiate it from other ambiguous classes. For some class $c$, to create non-differential descriptions, we first collect all the available differential descriptions, which is equal to the differential descriptions with $K = |C|$. Next, we create the normal set of differential descriptions, $D_c'$, with $K = 10$ as explained in Section 3. As a result of our description generation method, we know the corresponding attribute for each of the differential descriptions. Now, we filter the set of all available differential descriptions to exclude all the descriptions that their corresponding attribute is similar to the attributes explained by descriptions in $D_c'$. The remaining descriptions do not include any attribute that helps to separate the ambiguous classes.

We consider two attributes to be similar if they share a common word. For instance, `color` and `coat color` are similar since they share the word `color`. We split the attributes by white space and use simple string matching to check for this criteria. Because of the lack of diversity in the available descriptions for DTD, Oxford Pets, and Stanford Dogs datasets, using this criteria leads to a small number of remaining non-differential descriptions for each class. Therefore, for fair comparisons, we relax the criteria for these three datasets and compare attributes without splitting by white space, i.e., `color` and `coat color` are not considered similar for these three datasets.

As mentioned by previous work, the number of descriptions for each class also impacts the accuracy (Radford et al., 2021; Roth et al., 2023). We use description augmentation to control for the number of prompts for both differential and non-differential descriptions. Description augmentation creates a large number of descriptions from the original class descriptions with minimal impact on semantic information. Specifically, we create an augmented set of descriptions for each class description by replacing the original prefix (e.g. `a photo of a`) with a set of similar prefixes like `an image of a` and `a snapshot of a`. We use the prefixes used in Radford et al. (2021). Now to calculate the augmented description embedding, we average over the embeddings of all the descriptions in the corresponding augmented description set.

## D  COSTS

Although FuDD queries the LLM more than naive approaches, the API calls are very affordable and do not hinder wider adoption of FuDD. On average, one input prompt and model response

is 380 and 199 tokens, respectively. With OpenAI pricing at the time of writing ($0.001/1k and $0.002/1k tokens for input and response), the cost is $0.78 per 1000 queries, leading to affordable prices as reported in Table 7. FuDD also accommodates datasets with a large number of classes like ImageNet by recognizing the more significant role of ambiguous classes, reducing the costs for ImageNet dataset from $388 to $60 (see Appendix B for details).

In addition, as studied extensively in Section 5, we can use off-the-shelf or fine-tuned LLMs like Llama 2 to generate differential descriptions using in-house hardware to avoid API costs or accommodate other issues like working with private and sensitive data.

## E  ADDITIONAL VLMS

To further evaluate FuDD, we repeat our main experiments with different VLMs. We choose Open-CLIP (Cherti et al., 2023; Ilharco et al., 2021) because of its superior performance to CLIP. For example, OpenCLIP with ViT-L/14 backbone trained on `datacomp_xl_s13b_b90k` improves the performance of its CLIP counterpart by 4 percentage points on ImageNet dataset using Single Template descriptions. We also run experiments using BLIP-2 (Li et al., 2023) because it is trained using an entirely different strategy with a combination of image-text contrastive loss, image-text matching loss, and generation loss. To calculate image-text similarity using BLIP-2, we follow a similar procedure to Li et al. (2023). As reported in Table 8, these other VLMs can also use the additional information provided by FuDD and improve the performance beyond naive LLM-generated descriptions.

Table 8: FuDD accuracy using OpenCLIP and BLIP2 models. DComp is the checkpoint trained on `datacomp_xl_s13b_b90k` and Laion is the checkpoint trained on `laion2b_e16` dataset.

| Description | Cub | | | | DTD | | | |
| --- | --- | --- | --- | --- | --- | --- | --- | --- |
| | ViT-B-32 | | ViT-L-14 | BLIP2 | ViT-B-32 | | ViT-L-14 | BLIP2 |
| | DComp | Laion | DComp | - | DComp | Laion | DComp | - |
| Single Template | 72.63 | 63.93 | 85.31 | 23.52 | 54.95 | 51.12 | 63.19 | 53.24 |
| Template Set | 72.52 | 63.12 | 85.00 | 27.36 | 55.80 | 52.71 | 64.79 | 55.53 |
| Naive | 73.63 | 63.79 | 85.66 | 27.51 | **58.99** | **56.86** | 66.22 | 55.48 |
| FuDD ($k=10$) | 73.42 | 63.91 | 85.69 | 28.51 | 58.24 | 55.43 | 67.23 | 56.70 |
| FuDD ($k=|C|$) | **73.82** | **64.19** | **86.16** | **28.77** | 57.45 | 54.95 | **67.77** | **56.86** |

| | EuroSAT | | | | FGVCAircraft | | | |
| --- | --- | --- | --- | --- | --- | --- | --- | --- |
| | ViT-B-32 | | ViT-L-14 | BLIP2 | ViT-B-32 | | ViT-L-14 | BLIP2 |
| | DComp | Laion | DComp | - | DComp | Laion | DComp | - |
| Single Template | 38.03 | 41.73 | 61.14 | 52.14 | 29.94 | 26.31 | 51.82 | 14.19 |
| Template Set | 41.02 | 41.44 | 61.62 | 51.01 | 30.75 | 26.46 | 51.88 | 14.46 |
| Naive | 49.28 | 45.37 | 69.72 | 63.63 | 31.41 | 25.77 | **52.09** | 16.35 |
| FuDD ($k=10$) | **55.74** | **57.27** | **74.05** | **70.84** | **31.59** | **26.67** | 50.89 | 15.54 |
| FuDD ($k=|C|$) | **55.74** | **57.27** | **74.05** | **70.84** | 31.38 | 26.43 | 51.25 | **16.44** |

| | Flowers102 | | | | Food101 | | | |
| --- | --- | --- | --- | --- | --- | --- | --- | --- |
| | ViT-B-32 | | ViT-L-14 | BLIP2 | ViT-B-32 | | ViT-L-14 | BLIP2 |
| | DComp | Laion | DComp | - | DComp | Laion | DComp | - |
| Single Template | 72.13 | 67.49 | 80.60 | 55.91 | 85.89 | 81.31 | 94.49 | 85.07 |
| Template Set | 72.06 | 67.67 | 81.12 | 57.85 | 85.75 | 81.24 | 94.39 | 86.74 |
| Naive | 73.18 | 66.53 | 79.90 | 60.81 | 85.49 | 81.48 | 94.05 | 87.71 |
| FuDD ($k=10$) | 73.98 | 69.00 | **83.35** | **61.60** | 86.40 | 81.50 | 94.43 | 88.50 |
| FuDD ($k=|C|$) | **75.05** | **69.91** | 83.02 | 61.21 | **86.49** | **81.86** | **94.51** | **88.63** |

| | ImageNet | | | | ImageNet V2 | | | |
| --- | --- | --- | --- | --- | --- | --- | --- | --- |
| | ViT-B-32 | | ViT-L-14 | BLIP2 | ViT-B-32 | | ViT-L-14 | BLIP2 |
| | DComp | Laion | DComp | - | DComp | Laion | DComp | - |
| Single Template | 68.44 | 65.20 | 78.83 | 60.93 | 60.33 | 56.91 | 71.96 | 56.20 |
| Template Set | 69.13 | 65.61 | 79.15 | 66.07 | 60.76 | 57.36 | 72.05 | 60.60 |
| Naive | 68.60 | 65.42 | 79.03 | 66.15 | 60.06 | 57.19 | 71.92 | 61.00 |
| FuDD ($k=10$) | 69.13 | 65.91 | 79.25 | 67.31 | 60.94 | 57.70 | 72.08 | 61.28 |
| FuDD ($k=|C|$) | **69.35** | **66.20** | **79.50** | **68.55** | **61.28** | **57.90** | **72.39** | **62.53** |

| | Oxford Pets | | | | Places365 | | | |
| --- | --- | --- | --- | --- | --- | --- | --- | --- |
| | ViT-B-32 | | ViT-L-14 | BLIP2 | ViT-B-32 | | ViT-L-14 | BLIP2 |
| | DComp | Laion | DComp | - | DComp | Laion | DComp | - |
| Single Template | 89.40 | 87.54 | 94.74 | 76.70 | 41.52 | 41.88 | 43.21 | 43.72 |
| Template Set | 88.47 | 87.49 | 93.51 | 76.91 | **43.20** | 42.84 | 44.73 | 43.67 |
| Naive | 89.86 | 89.07 | 94.79 | 81.17 | 42.24 | 42.61 | 44.06 | 43.51 |
| FuDD ($k=10$) | 90.71 | 89.04 | 94.90 | 81.96 | 42.79 | 43.16 | **44.98** | 45.30 |
| FuDD ($k=|C|$) | **90.95** | **90.00** | **95.18** | **83.51** | 43.13 | **43.55** | 44.89 | **45.52** |

| | Stanford Cars | | | | Stanford Dogs | | | |
| --- | --- | --- | --- | --- | --- | --- | --- | --- |
| | ViT-B-32 | | ViT-L-14 | BLIP2 | ViT-B-32 | | ViT-L-14 | BLIP2 |
| | DComp | Laion | DComp | - | DComp | Laion | DComp | - |
| Single Template | 88.42 | 86.82 | 93.67 | 79.97 | 63.92 | 59.76 | 79.23 | 47.70 |
| Template Set | **88.66** | **87.02** | 93.71 | 80.48 | 64.93 | 59.50 | 79.22 | 47.76 |
| Naive | 87.38 | 86.76 | 93.56 | 80.08 | 65.02 | 60.12 | 79.22 | 50.31 |
| FuDD ($k=10$) | 88.35 | 86.87 | **93.77** | 80.25 | 65.13 | 59.84 | 79.39 | 52.42 |
| FuDD ($k=|C|$) | 88.45 | 86.84 | 93.65 | **81.06** | **65.96** | **60.22** | **80.29** | **52.91** |

| Class 1 | Class 0: Tennessee Warbler  |
|---|---|
| Black-footed Albatross  | Attribute: size
0: A photo of a tennessee warbler, a small songbird that is only about 4 inches long.
1: A photo of a black-footed albatross, a large seabird with a wingspan of up to 7 feet.

Attribute: coloration
0: A photo of a tennessee warbler, a bright yellow bird with olive-green wings and back.
1: A photo of a black-footed albatross, a dark-colored bird with a white head and underparts.

Attribute: bill shape
0: A photo of a tennessee warbler, a bird with a small, pointed bill.
1: A photo of a black-footed albatross, a bird with a large, hooked bill. |
| Mangrove Cuckoo  | Attribute: bill length
0: A photograph of a tennessee warbler, a type of bird, with a short, pointed bill.
1: A photograph of a mangrove cuckoo, a type of bird, with a long, curved bill.

Attribute: tail length
0: A photograph of a tennessee warbler, a type of bird, with a short, square tail.
1: A photograph of a mangrove cuckoo, a type of bird, with a long, graduated tail.

Attribute: wing bars
0: A photograph of a tennessee warbler, a type of bird, with two white wing bars.
1: A photograph of a mangrove cuckoo, a type of bird, with no wing bars. |
| Caspian Tern  | Attribute: size
0: A photograph of a tennessee warbler, a type of bird, which is smaller in size.
1: A photograph of a caspian tern, a type of bird, which is larger in size.

Attribute: bill color
0: A photograph of a tennessee warbler, a type of bird, with a black bill.
1: A photograph of a caspian tern, a type of bird, with a red bill.

Attribute: wing shape
0: A photograph of a tennessee warbler, a type of bird, with short, rounded wings.
1: A photograph of a caspian tern, a type of bird, with long, pointed wings. |
| Black-throated Sparrow  | Attribute: head pattern
0: A photograph of a tennessee warbler, a type of bird, with yellow and gray head pattern.
1: A photograph of a black-throated sparrow, a type of bird, with black and white striped head pattern.

Attribute: eye ring
0: A photograph of a tennessee warbler, a type of bird, with no eye ring.
1: A photograph of a black-throated sparrow, a type of bird, with white eye ring.

Attribute: breast color
0: A photograph of a tennessee warbler, a type of bird, with yellow breast color.
1: A photograph of a black-throated sparrow, a type of bird, with black and white striped breast color. |

Figure 7: Various pairwise differential descriptions for sample class "Tennessee Warbler" and four different classes.

Table 9: Prompt template for generating pairwise differential descriptions. We use two fixed sample pairs of classes to guide the model to generate the appropriate output.

| Type | Content |
|---|---|
| System | You are a knowledgeable teacher. |
| User | For the following objects, generate captions that represent the distinguishing visual differences between the photos of the two objects.
Generate as many captions as you can.

Object 1:  Field sparrow
Object 2:  Acadian flycatcher |
| Assistant | <START>
Visual characteristic:  Bill shape
Caption 1:  A photograph of a field sparrow, a type of bird, with cone bill shape.
Caption 2:  A photograph of a acadian flycatcher, a type of bird, with all-purpose bill shape.

Visual characteristic:  Breast color
Caption 1:  A photograph of a field sparrow, a type of bird, with buff breast color.
Caption 2:  A photograph of a acadian flycatcher, a type of bird, with white breast color.

Visual characteristic:  Back pattern
Caption 1:  A photo of a field sparrow, a type of bird, with striped back pattern.
Caption 2:  A photo of a acadian flycatcher, a type of bird, with solid back pattern.
<END> |
| User | For the following objects, generate captions that represent the distinguishing visual differences between the photos of the two objects.
Generate as many captions as you can.

Object 1:  Cornet
Object 2:  Flute |
| Assistant | <START>
Visual characteristic:  Shape
Caption 1:  A photo of a cornet, a type of musical instrument, with a conical bore.
Caption 2:  A photo of a flute, a type of musical instrument, with a cylindrical bore.
<END> |
| User | For the following objects, generate captions that represent the distinguishing visual differences between the photos of the two objects.
Generate as many captions as you can.

Object 1:  {class name 1}
Object 2:  {class name 2} |

Table 10: Sample descriptions generated by GPT-3.5 for Cub

| Classes | Descriptions |
|---------|--------------|
| 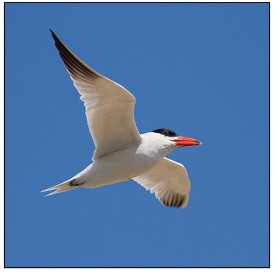 Caspian Tern 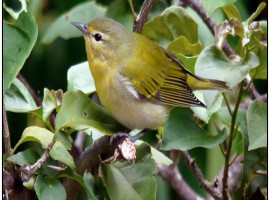 Tennessee Warbler | Attribute:  size
0:  a photograph of a caspian tern, a type of bird, which is larger in size.
1:  a photograph of a tennessee warbler, a type of bird, which is smaller in size.

Attribute:  bill color
0:  a photograph of a caspian tern, a type of bird, with a red bill.
1:  a photograph of a tennessee warbler, a type of bird, with a black bill.

Attribute:  wing shape
0:  a photograph of a caspian tern, a type of bird, with long, pointed wings.
1:  a photograph of a tennessee warbler, a type of bird, with short, rounded wings.

Attribute:  tail shape
0:  a photograph of a caspian tern, a type of bird, with a forked tail.
1:  a photograph of a tennessee warbler, a type of bird, with a square tail. |

Table 11: Sample descriptions generated by GPT-3.5 for DTD

| Classes | Descriptions |
|---------|--------------|
| 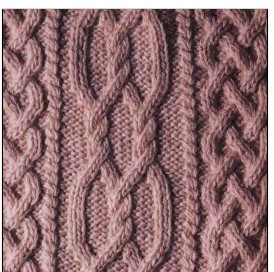 Interlaced 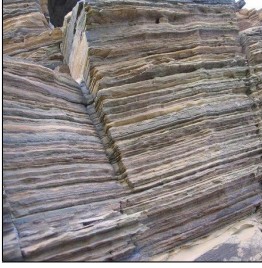 Stratified | Attribute:  texture
0:  a photo of an interlaced surface, with a woven texture.
1:  a photo of a stratified surface, with a layered texture.

Attribute:  pattern
0:  a photo of an interlaced surface, with a criss-cross pattern.
1:  a photo of a stratified surface, with a horizontal pattern.

Attribute:  material
0:  a photo of an interlaced surface, made of woven fibers.
1:  a photo of a stratified surface, made of layered sediment or rock. |

Table 12: Sample descriptions generated by GPT-3.5 for EuroSAT

| Classes | Descriptions |
|---|---|
| 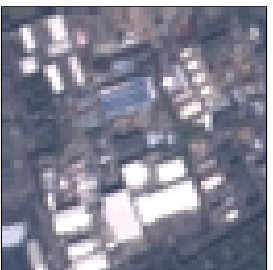
Industrial or commercial building

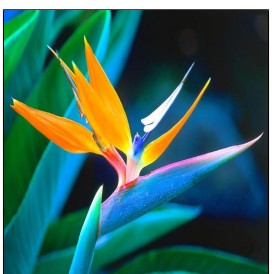
River | `Attribute: color`
`0: a satellite image of industrial or commercial buildings, with a mix of grey, white, and black colors.`
`1: a satellite image of a river, with blue and green colors.`

`Attribute: texture`
`0: a satellite image of industrial or commercial buildings, with a rough and angular texture.`
`1: a satellite image of a river, with a smooth and flowing texture.`

`Attribute: shape`
`0: a satellite image of industrial or commercial buildings, with rectangular and square shapes.`
`1: a satellite image of a river, with a winding and curvy shape.`

`Attribute: pattern`
`0: a satellite image of industrial or commercial buildings, with a grid-like pattern.`
`1: a satellite image of a river, with a meandering pattern.` |

Table 13: Sample descriptions generated by GPT-3.5 for Flowers

| Classes | Descriptions |
|---|---|
| 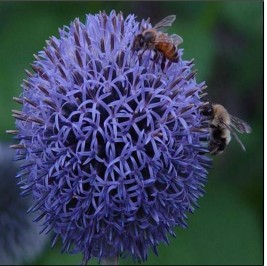
Bird of paradise

Globe thistle | `Attribute: color`
`0: a photo of a bird of paradise, a type of flower, with bright orange and blue colors.`
`1: a photo of a globe thistle, a type of flower, with muted blue and green colors.`

`Attribute: shape`
`0: a photo of a bird of paradise, a type of flower, with a unique bird-like shape.`
`1: a photo of a globe thistle, a type of flower, with a spherical shape.`

`Attribute: texture`
`0: a photo of a bird of paradise, a type of flower, with smooth and glossy petals.`
`1: a photo of a globe thistle, a type of flower, with spiky and rough texture.` |

Table 14: Sample descriptions generated by GPT-3.5 for Food101

| Classes | Descriptions |
|---|---|
| 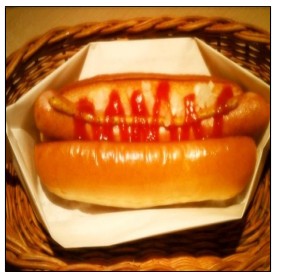
Hot dog

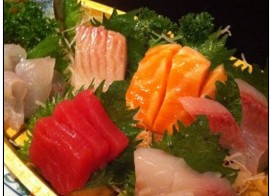
Sashimi | Attribute:  type of food
0:  a photo of a hot dog, a type of fast food, with a sausage in a bun.
1:  a photo of sashimi, a type of japanese cuisine, with raw fish slices.

Attribute:  cooking method
0:  a photo of a hot dog, a type of fast food, with a grilled sausage.
1:  a photo of sashimi, a type of japanese cuisine, with raw fish slices.

Attribute:  serving style
0:  a photo of a hot dog, a type of fast food, served with ketchup and mustard.
1:  a photo of sashimi, a type of japanese cuisine, served with soy sauce and wasabi.

Attribute:  texture
0:  a photo of a hot dog, a type of fast food, with a chewy texture.
1:  a photo of sashimi, a type of japanese cuisine, with a soft and tender texture. |

Table 15: Sample descriptions generated by GPT-3.5 for ImageNet

| Classes | Descriptions |
|---|---|
| 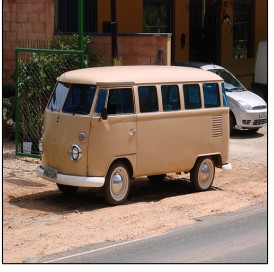
Minivan

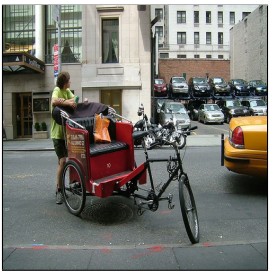
Rickshaw | Attribute:  number of wheels
0:  a photo of a minivan, which has four wheels.
1:  a photo of a rickshaw, which has three wheels.

Attribute:  size
0:  a photo of a minivan, which is larger in size and can accommodate more passengers.
1:  a photo of a rickshaw, which is smaller in size and can accommodate fewer passengers.

Attribute:  propulsion
0:  a photo of a minivan, which is powered by an engine.
1:  a photo of a rickshaw, which is powered by human pedaling.

Attribute:  type of vehicle
0:  a photo of a minivan, which is a modern automobile.
1:  a photo of a rickshaw, which is a traditional asian vehicle. |

Table 16: Sample descriptions generated by GPT-3.5 for Oxford Pets

| Classes | Descriptions |
|---------|--------------|
| 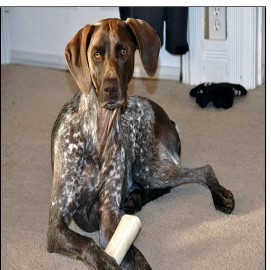 German shorthaired 

 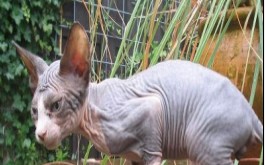 Sphynx | ```Attribute:  body type 0:  a photo of a german shorthaired, a type of dog, with a muscular and athletic body type. 1:  a photo of a sphynx, a type of cat, with a slender and sleek body type.  Attribute:  coat 0:  a photo of a german shorthaired, a type of dog, with a short and dense coat. 1:  a photo of a sphynx, a type of cat, with no coat or hair.  Attribute:  ears 0:  a photo of a german shorthaired, a type of dog, with floppy ears. 1:  a photo of a sphynx, a type of cat, with large and pointed ears.  Attribute:  facial features 0:  a photo of a german shorthaired, a type of dog, with a snout and a prominent nose. 1:  a photo of a sphynx, a type of cat, with a flat face and no visible nose bridge.``` |

Table 17: Sample descriptions generated by GPT-3.5 for Stanford Cars

| Classes | Descriptions |
|---------|--------------|
| 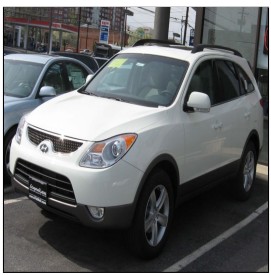 2012 Hyundai Veracruz SUV 

 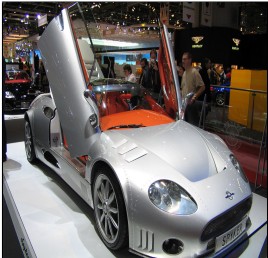 2009 Spyker C8 Convertible | ```Attribute:  body type 0:  a photo of a 2012 hyundai veracruz suv, a large vehicle with a high ground clearance and a boxy shape. 1:  a photo of a 2009 spyker c8 convertible, a sleek and low-slung sports car with a convertible top.  Attribute:  number of doors 0:  a photo of a 2012 hyundai veracruz suv, a vehicle with four doors. 1:  a photo of a 2009 spyker c8 convertible, a vehicle with two doors.  Attribute:  wheel design 0:  a photo of a 2012 hyundai veracruz suv, with standard alloy wheels. 1:  a photo of a 2009 spyker c8 convertible, with unique and intricate spoke wheels.  Attribute:  grille design 0:  a photo of a 2012 hyundai veracruz suv, with a large and prominent grille. 1:  a photo of a 2009 spyker c8 convertible, with a small and distinctive grille.``` |

Table 18: Sample descriptions generated by GPT-3.5 for Stanford Dogs

| Classes | Descriptions |
|---------|--------------|
| 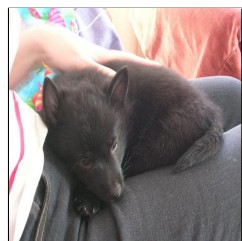 Schipperke 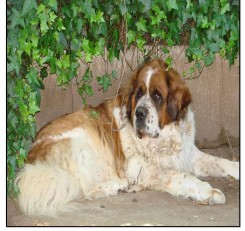 Saint bernard | ```
Attribute:  size
0:  a photo of a schipperke dog, a small breed of
dog.
1:  a photo of a saint bernard dog, a large breed
of dog.

Attribute:  coat color
0:  a photo of a schipperke dog, with black coat
color.
1:  a photo of a saint bernard dog, with white and
brown coat color.

Attribute:  ear shape
0:  a photo of a schipperke dog, with pointed ears.
1:  a photo of a saint bernard dog, with droopy
ears.

Attribute:  tail length
0:  a photo of a schipperke dog, with a short tail.
1:  a photo of a saint bernard dog, with a long
tail.
``` |

Table 19: Sample descriptions generated by GPT-3.5 for FGVC Aircraft

| Classes | Descriptions |
|---------|--------------|
| 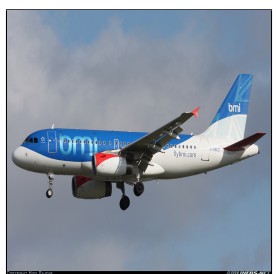 Airbus A319 Cessna 172 | ```
Attribute:  size
0:  a photo of an airbus a319 aircraft, a
commercial airliner that is much larger than a
cessna 172.
1:  a photo of a cessna 172 aircraft, a small,
single-engine plane that is much smaller than an
airbus a319.

Attribute:  wing shape
0:  a photo of an airbus a319 aircraft, with
swept-back wings.
1:  a photo of a cessna 172 aircraft, with
straight wings.

Attribute:  engine placement
0:  a photo of an airbus a319 aircraft, with
engines mounted under the wings.
1:  a photo of a cessna 172 aircraft, with a
single engine mounted on the nose of the plane.

Attribute:  cockpit windows
0:  a photo of an airbus a319 aircraft, with a
large cockpit window that extends over the top of
the plane.
1:  a photo of a cessna 172 aircraft, with a
small, single-piece windshield in the cockpit.
``` |

Table 20: Sample descriptions generated by GPT-3.5 for Places365

| Classes | Descriptions |
|---|---|
| 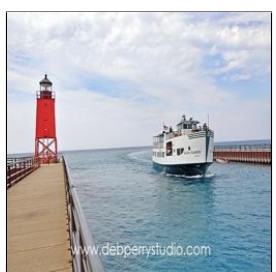
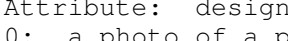
Pier

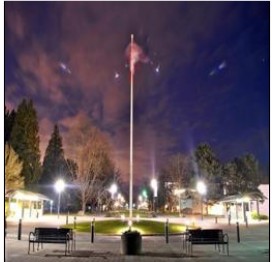
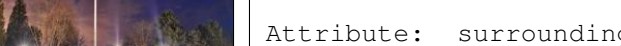
Plaza | Attribute:  location
0:  a photo of a pier, which is located near a body of water.
1:  a photo of a plaza, which is located in a city or town center.

Attribute:  purpose
0:  a photo of a pier, which is used for docking boats and ships.
1:  a photo of a plaza, which is used for public gatherings and events.

Attribute:  design
0:  a photo of a pier, which is typically long and narrow with a flat surface.
1:  a photo of a plaza, which is typically open and spacious with various features like fountains, benches, and sculptures.

Attribute:  surroundings
0:  a photo of a pier, which is surrounded by water and may have views of the ocean or other bodies of water.
1:  a photo of a plaza, which is surrounded by buildings and may have views of city streets and architecture. |

