# OpenReview forum: "Follow-Up Differential Descriptions: Language Models Resolve Ambiguities for Image Classification"
_ICLR.cc/2024/Conference — ICLR 2024 poster_

### Official Review · Reviewer_LDou · 2023-10-23

**Soundness:** 3 good
**Presentation:** 3 good
**Contribution:** 3 good
**Rating:** 6
**Confidence:** 3

**Summary:**

This paper studies how to extend the class descriptions to improve the performance of CLIP-style vision-language models (VLMs) in image classification. It points out that the class descriptions generated by existing methods may not provide useful information that differentiates the class from other classes.

To address this problem, this paper proposes the Follow-up Differential Descriptions (FuDD). FuDD first identifies a set of ambiguous classes using the basic class descriptions. Then, for each ambiguous class, a LLM is prompted to generate descriptions, which involve information that can distinguish it from other ambiguous classes. The generated class descriptions are then used to obtain the final classification results.

The empirical studies show that FuDD outperforms the basic class descriptions and naïve LLM generated descriptions that do not consider differential attributes between classes. The experiments include two kinds of LLMs, i.e., the GPT-3.5 (accessed via API) and the open-sourced LLAMa2. It is shown that LLAMa2-based FuDD can achieve effective results after being fine-tuned on the descriptions generated by GPT-3.5.

**Strengths:**

* The idea of generating differential class descriptions is well-motivated and reasonable.
* The experiments are well-designed and the results are convincing, which demonstrates the effectiveness of using differential class descriptions.
* The paper is well-written and easy to follow.

**Weaknesses:**

* The experiments only include one type of VLM, i.e., CLIP. The reviewer would like to know whether FuDD is compatible with more advanced VLMs.
* There is no analysis on the trade-off between performance gain by FuDD and additional (computational or financial) cost. Since generating pair-wise class description is more expensive than generating independent description for each class, such analysis is important.

**Questions:**

N/A

---

> ### Author Response · Authors · 2023-11-20
> **Response to reviewer LDou**
>
> Thank you for your thoughtful comments on our manuscript. Below we address your comments and questions about our work.
>
> > The experiments only include one type of VLM, i.e., CLIP. The reviewer would like to know whether FuDD is compatible with more advanced VLMs.
>
> Thank you for your suggestion. To further evaluate FuDD, we run additional experiments with OpenCLIP, which is much stronger than CLIP, and BLIP2, which uses an entirely different training strategy. We observe similar performance improvement in this new set of experiments, which further supports the argument that FuDD benefits a wide range of VLMs.
> Please see our general comment to all reviewers for the results and further discussion.
>
>
> > There is no analysis on the trade-off between performance gain by FuDD and additional (computational or financial) cost. Since generating pair-wise class description is more expensive than generating independent description for each class, such analysis is important.
>
> The number of LLM queries directly depends on the number of ambiguous classes, which is a hyperparameter set by the user. In Figure 3, we report the performance gains with different numbers of ambiguous classes (i.e., different values for k). As discussed in Section 4.1, the five most ambiguous classes account for most of FuDD’s performance gains. As a result, to further decrease the costs, one can use smaller values for k, while preserving most of FuDD’s benefits.
> Please also see our general comment to all reviewers for exact API costs and further discussion.

---

> > ### Comment · Reviewer_LDou · 2023-11-22
> > **Response to Author Rebuttal**
> >
> > Thanks for taking the time to respond to my comments. It is nice to see that (1) FuDD can also boost the performance of OpenCLIP and BLIP2 and (2) the information on API costs, which enhances the soundness of the paper.

---

### Official Review · Reviewer_Y5Zb · 2023-11-01

**Soundness:** 3 good
**Presentation:** 3 good
**Contribution:** 3 good
**Rating:** 8
**Confidence:** 4

**Summary:**

This paper studies the prompt engineering. Instead of directly asking LLM or designing a set of prompts, the author proposed to let the model decides which sets of categories are ambiguous and then ask the LLM to clarify those sets of categories. This is actually a quite interesting and smart idea. Comparing with the other prompt engineering approach, the proposed one achieves higher performance.

**Strengths:**

1. The proposed idea makes a lot of sense and achieves quite impressive performance.

2. The paper showed that by finetuning the open-sourced model, the open-sourced model (LLAMA2) could also generate the descriptions with discriminative capability across ambiguous categories.

3. The proposed approach achieves much better performance than simple prompt engineering.

4. If this is the first paper presented in this field, I would rate a strong acceptance. This is a little bit out of my domain, I would rate acceptance.

**Weaknesses:**

I think this paper is interesting and the proposed approaches make sense. I would not be able to state the weaknesses.

**Questions:**

1. The section 5: I wonder why would the Llama2 can generate informative sentences for satellite images? Especially given the model only been finetuned on the ImageNet dataset sentences.

2. What if the caption is so long and so complex that the image-text model could not understand? I wonder if the author encountered this scenario, if so, how to address it?

---

> ### Author Response · Authors · 2023-11-20
> **Response to reviewer Y5Zb**
>
> Thank you for your thoughtful comments on our manuscript. Below we address your comments and questions about our work.
>
> > If this is the first paper presented in this field, I would rate a strong acceptance.
>
> Our work is the first to propose a **zero-shot**, **training-free** approach for generating class descriptions that better **differentiate** between the target classes, as well as providing analysis of what the characteristics of such descriptions are. Previous zero-shot approaches dismiss the commonalities between classes, and the closest work that aims for differentiating descriptions is LaBo[1] which uses training data and further optimization to select a subset of naive class descriptions that are more differentiating.
>
> > The section 5: I wonder why would the Llama2 can generate informative sentences for satellite images? Especially given the model only been finetuned on the ImageNet dataset sentences.
>
> We believe that during fine-tuning, the LLM learns what types of features are useful for differentiating the target classes, while the world knowledge comes from the pre-training stage. As shown in Figure 4 and Figure 5, after fine-tuning, the LLM is describing more low-level visual features instead of higher-level semantic concepts. As another piece of evidence in support of this argument, in Table 3, we observe that the correctness of the descriptions increases moderately (i.e., fine-tuning has limited impact on LLM’s world knowledge), while their helpfulness (i.e., being differentiating) increases significantly, which we attribute the increased performance to.
>
> > What if the caption is so long and so complex that the image-text model could not understand? I wonder if the author encountered this scenario, if so, how to address it?
>
> That was exactly the case in our early experiments, where we got long, complex descriptions, sometimes expanding an entire paragraph. To alleviate this problem, we take advantage of the in-context learning capabilities of LLMs, and use two examples as part of our LLM prompt to guide the model to focus on concise visual features and follow the description style that is known to work well for VLMs, i.e., short photo captions that start with prefixes like "a photo of a."
>
> Inspecting the generated descriptions across all datasets, we find that, on average, each description contains 14 words and 99 percent of the descriptions start with simple prefixes like:
> "a photo of"
> "a photograph of"
> "a video of"
> "a close-up of"
> "a screenshot of"
> "a close-up photo of"
> "a microscopic image of"
>
> Excluding the prefix and the class name, there are approximately 10 remaining words in each description, which does not allow for much complexity in descriptions.
>
> Moreover, to keep the descriptions simple but at the same time benefit from a diverse set of described attributes, we guide the model in both the task instruction and the in-context examples to generate multiple descriptions, and then we average over the generated descriptions to compute the class representations. On average, over all datasets, we use approximately 4 descriptions per class.
>
> [1] Yang, Y., Panagopoulou, A., Zhou, S., Jin, D., Callison-Burch, C., & Yatskar, M. (2023). Language in a bottle: Language model guided concept bottlenecks for interpretable image classification. In CVPR 2023.

---

### Official Review · Reviewer_sQgp · 2023-11-01

**Soundness:** 3 good
**Presentation:** 3 good
**Contribution:** 2 fair
**Rating:** 5
**Confidence:** 4

**Summary:**

This paper proposed a simple data augmentation technique for zero-shot image classification tasks. Specifically, the idea is to utilize large language models (LLM) to generate more descriptive and informative class labels for ambiguous classes, with the hope that they can help the base image-text retrieval/classification model to classify the image. Experimental results suggest promising gains over baseline, in this case a CLIP model without this augmentation technique.

**Strengths:**

1. The proposed method is intuitive. It is expected that using more descriptive class labels can improve zero-shot tasks.
2. Experimental results show consistent gains across different datasets.

**Weaknesses:**

1. The method is incremental, as it is a small trick to boost zero-shot image classification results.
2. The baseline is not strong enough. It is therefore not clear if the method can still improve state-of-the-art models on these tasks.
3. More diverse and challenging tasks are missing from the experiment, e.g. image-text retrieval. It will be nice to show wider applicability of the method.

**Questions:**

NA

---

> ### Author Response · Authors · 2023-11-20
> **Response to reviewer sQgp**
>
> Thank you for your thoughtful comments on our manuscript. Below we address your comments and questions about our work.
>
> > The method is incremental, as it is a small trick to boost zero-shot image classification results.
>
> We believe that our work contributes significantly to the field, especially as the need for efficient approaches to adapting VLMs to downstream tasks increases. We propose technical contributions that outperform previous methods, provide analysis of characteristics of effective natural language descriptions, study the utility of open-source LLMs and their fine-tuning,  and contribute to answering open questions in the field on the potential of natural language class descriptions for VLM adaptation compared to other sources of information (e.g., synthetic images) or alternative approaches like soft prompt tuning.
>
> Please see our general comment to all reviewers for an extended discussion on our contributions.
>
> > The baseline is not strong enough. It is therefore not clear if the method can still improve state-of-the-art models on these tasks.
>
> To address the reviewer’s concern, we run additional experiments on two OpenCLIP model variants, which are stronger than their CLIP counterpart, as well as the BLIP2 model, which uses a different style of pre-training than CLIP. In both cases, we observe similar improvements, where the additional information provided by FuDD improves the performance over the naive LLM-generated descriptions.
>
> Please see our general comment to all reviewers for additional experiments and further discussion.
>
> > More diverse and challenging tasks are missing from the experiment, e.g. image-text retrieval. It will be nice to show wider applicability of the method.
>
> There are many interesting vision and language tasks that could benefit from additional semantic information. However, in keeping with most of the closely related work, here, we also focus on image classification, which remains a challenging and important problem (please refer to the manuscript for an extended discussion of previous work). Focusing on image classification allows more diverse comparisons and insights in this line of work, without confounding factors like task variation. For example, the comparison of image classification results in Section 6 uncovers the potential of natural language descriptions for VLM adaptation compared to other modalities or few-shot approaches proposed by previous work.
>
> Although outside of the focus of our work, we would like to discuss FuDD in the context of information retrieval (IR). We draw an analogy between FuDD and reranking methods for IR, where for each query, a low-cost approach selects a subset of potentially related documents (similar to Section 3.2 in FuDD), and a more complex method reranks this smaller subset of documents (similar to Section 3.3 and 3.4 in FuDD). In the context of IR, because of the large number of documents, it is financially and computationally impossible to generate alternative descriptions for all documents regardless of choosing naive or differential descriptions. In such a setting, focusing on ambiguous documents is the only feasible option (we have extended discussion and experiments on ambiguous classes in Sections 3.2 and 4.1).
> Now the question is, "How effective are naive descriptions for reranking compared to FuDD?." Constrained by the rebuttal time window, we design new experiments to simulate text retrieval with our current descriptions and datasets. In these new experiments, we use test images as equivalent for queries and classes as the documents to be retrieved in a text-retrieval setup.
> We first retrieve the ambiguous classes as described in Section 3.2 and then use differential descriptions and naive descriptions to rerank these ambiguous classes and compare the results. We report the accuracy and nDCG@10, which represents the improvement in the ranking of the documents (i.e., class names). Compared to classification, it shows how much closer the true class label is to the top of the list, even if it is not the top prediction.
>
>
> |-|Flowers102|Flowers102|Places365|Places365|
> |---|---|---|---|---|
> |-|Acc|nDCG@10|ACC|nDCG@10|
> |Naive|75.98|85.21|41.58|59.37|
> |FuDD|78.76|86.35|43.95|60.89|
>
>
> As expected, we observe that better representations for each class coming from differential descriptions actually improve the performance in this case, where both naive and differential descriptions are used for reranking.

---

### Official Review · Reviewer_RXgc · 2023-11-01

**Soundness:** 3 good
**Presentation:** 3 good
**Contribution:** 2 fair
**Rating:** 5
**Confidence:** 4

**Summary:**

Recent work shows that evaluating vision-language models with descriptions on object attributes improves image classification performance. This work proposes to prompt LLMs to specifically give “differential” descriptions (i.e., distinctive attributes between ambiguous classes) and using such more targeted descriptions outperforms generic descriptions. The method shows performance improvement over the baselines on a range of image classification benchmarks.

After rebuttal: I appreciate the author's rebuttal and have raised my score to 5. However, I still believe that the contribution seems thin given that only a prompting technique is proposed.  In addition, compared to the baseline, the performance improvement is oftentimes not significant (Table 1); thus the generalization ability of the method is not very clear.

**Strengths:**

The idea is simple and natural. Directly prompting LLM to give attributes for a certain class is an underspecified task; there are so many attributes of different granularities to enumerate. Generating differential descriptions given two classes is a smart way to generate targeted attributes.

**Weaknesses:**

**1. No significant method improvement**

The idea of prompting LLM to give more differentiating descriptions is effective but not necessarily a significant innovation.


**2. Reliance on pre-trained vision-language models**

The paper focuses only on getting differential descriptions; however, it is not guaranteed that CLIP can accurately interpret them. The effectiveness of such models hinges critically on and is bounded by the vision-language models. I suspect that this could be why the performance improvement is not very significant and why different CLIP variants could exhibit huge performance variations.

I would appreciate an error analysis: how many of the mistakes are due to insufficient / erroneous differential descriptions and how many of the mistakes are due to CLIP’s insensitivity to descriptions? This could be informative for future work.



**3. API cost for a single inference on an image**

It is not immediately clear how many times we need to query LLM for a single inference on one image. For k=10, does it mean we need to create 55 queries? For experiments where k = |C|, will the cost become prohibitive (even with caching)?

It would be better if such details are discussed in the main text.

**Questions:**

- For the equation in 3.2, what does the summation mean? I guess the equation simply means getting the top-k classes with the highest similarity with the image features? Then in this case, do we need to sum over c_i?

- Would it be possible to give examples of differential descriptions for each of the evaluation datasets? It is especially interesting to see the huge performance on EuroSAT.

- Would it be possible to prompt the LLM to come up with differential attributes given >2 classes in a single prompt?

---

> ### Author Response · Authors · 2023-11-20
> **Response to reviewer RXgc**
>
> Thank you for your thoughtful comments on our manuscript. Below we address your comments and questions about our work.
>
> > No significant method improvement
>
> We respectfully disagree with the reviewer as our work makes technical contributions, achieves superior results, provides analysis on characteristics of effective natural language descriptions, studies the use of off-the-shelf and fine-tuned LLMs in the process, and positions natural language descriptions as a promising approach for adapting VLMs. This is especially important for future work as it advances the debate over promising approaches for VLM adaptation in favor of natural language descriptions compared to other sources of information like synthetic images or alternative approaches like soft prompt tuning.
> Please find our extended response in the general comment to all reviewers for further discussion.
>
> > The paper focuses only on getting differential descriptions; however, it is not guaranteed that CLIP can accurately interpret them. The effectiveness of such models hinges critically on and is bounded by the vision-language models.
>
> Here, we focus on generating class descriptions that provide the most useful information for the target set of classes and how we can use LLMs to do so effectively and efficiently. Of course, different VLMs use this information differently based on their capabilities and pre-training datasets. However, as shown in our experiments, VLMs are capable of taking advantage of this information to improve performance.
> Please see our general comment to all reviewers for experiments with additional VLMs, where we observe similar improvements.
> Clearly, VLMs are able to take advantage of this information. However, we believe an additional attempt to improve VLMs’ understanding of natural language is a different question and is out of the scope of this work.
>
> > [...] I suspect that this could be why the performance improvement is not very significant.
>
> Respectfully, considering the challenging nature of the task, we believe our method makes significant improvements in most cases (e.g., 13.95 and 5.58 percentage points for EuroSAT and Oxford Pets). Especially, considering that without using any labeled data, FuDD’s performance as a zero-shot method is comparable to other few-shot learning methods that use labeled images for adapting VLMs.
>
> > [...] how many of the mistakes are due to insufficient / erroneous differential descriptions and how many of the mistakes are due to CLIP’s insensitivity to descriptions?
>
> As an attempt to understand how the correctness and helpfulness (i.e., being differentiating) of descriptions impact the performance, we perform a manual evaluation as reported in Table 3, and observe that improving the helpfulness has a significant impact on performance even if the level of correctness of descriptions stays roughly the same. Another related experiment is described in Section 4.2, where we show that differentiating attributes are superior to generic attributes and that VLMs are sensitive to the described differences and can use them to their advantage.
>
> As explained in Section 3.3, we use an ensemble of descriptions for each class, and often the ensemble is a mixture of helpful and correct and unhelpful or incorrect descriptions. As a result, it is challenging, if at all possible, to decide if the wrong prediction is caused by incorrect or unhelpful information or by the model’s inability to understand the correct and helpful descriptions. Therefore, this hinders such a precise analysis of the cause of the error for each image as suggested by the reviewer.
>
> > API cost for a single inference on an image
>
> Please, find our response in the general comment to all reviewers.
>
> > For the equation in 3.2, what does the summation mean? [...]
>
> As mentioned by the reviewer, this equation selects the k-most similar classes to each image. In other words, the argmax is over sets of classes, and to score a set, we have to sum the similarities of all classes in each set.
>
> > Would it be possible to give examples of differential descriptions for each of the evaluation datasets?
>
> Thank you for your suggestion. We include examples of differential descriptions for all datasets in the updated manuscript. It is interesting to see that differential descriptions are more effective in cases that describe low-level visual characteristics than high-level semantic features, which is consistent with our previous observations described in Section 5.
>
> > Would it be possible to prompt the LLM to come up with differential attributes given >2 classes in a single prompt?
>
> We think it is really interesting to investigate these different approaches, and we do not see an immediate reason that it does not work. Since this is the first work on generating differentiating, dataset-specific class descriptions, we focused on two classes per query. But, this is an interesting direction for future work.

---

### Author Response · Authors · 2023-11-20
**To all reviewers (1/2)**

## Significance of Contributions

We would like to reiterate our contributions in response to reviewer RXgc and sQgp’s concerns about the significance of our work.
From a technical perspective:
- To the best of our knowledge, we are the first zero-shot approach to consider the interactions between target classes for VLM adaptation without using any labeled data or further optimization.
- As noted by all reviewers, through extensive experiments, we show that our method outperforms previous work on 12 datasets with different visual backbones.
- We design further analytical experiments to systematically study the value of differentiating attributes. We find that differentiating attributes are clearly superior to generic attributes, and importantly, VLMs are sensitive to this information and can use it to their advantage.
- Through rigorous experiments, we show that ambiguous classes for each image are the main culprit for image classification that we should focus on, enabling us to design an efficient approach for dealing with large numbers of classes.
- In addition to proprietary LLMs like ChatGPT, we investigate the use of open-source LLMs like Llama2. We study the impact of further fine-tuning on the performance and structure of prompts.

Regarding the novelty of our work, we quote reviewer Y5Zb: "[...] the author proposed to let the model decides which sets of categories are ambiguous and then ask the LLM to clarify those sets of categories", and in their opinion  "This is actually a quite interesting and smart idea."

We would also like to discuss the contributions of our work in the context of recently published papers in this area. After an initial burst of activity in adapting VLMs through natural language descriptions, the most recent works in this area have moved towards other sources of information like labeled examples and synthetic images. Our work shows that smart approaches for generating natural language descriptions, even in a zero-shot setup, can still match or outperform these more complex alternatives.
- There is a growing effort to adapt VLMs to downstream tasks through natural language descriptions due to their efficiency (e.g., [1] and [2] in  ICLR 2023 and ICML 2023 ). Our work clearly advances this effort by introducing FuDD, achieving superior performance, and providing analysis of characteristics of effective descriptions.
- Appreciating the importance of differentiating information, LaBo [3] (CVPR 2023) uses labeled images and further optimization to find such differentiating descriptions. On the other hand, we propose a zero-shot, training-free approach to achieve this goal and obtain competitive results.
- Trying to evaluate the value of semantic information, WaffleCLIP[4] (ICCV 2023) casts doubt on the value of additional semantic information for image classification. However, we show that additional semantic information could significantly improve the performance if chosen appropriately, as done in FuDD.
- SuS-X [5] (ICCV 2023) suggests that natural language descriptions provide limited information, and it is better to use them to generate synthetic labeled images as a more effective modality. On the contrary, we show that natural language descriptions are as effective if chosen appropriately as done in FuDD and achieve comparable performance while avoiding the complexities of text-to-image generation models.
- In general, as discussed in section 6, our work establishes natural language descriptions as an efficient approach for VLM adaptation, without the complexities of its alternatives, while maintaining competitive performance.

To emphasize the motivations for our work, we quote reviewer LDou: "The idea of generating differential class descriptions is well-motivated and reasonable."

Finally, we see the simplicity not as a disadvantage, but as a strength of our work that allows further applicability and adoption of our findings.

References

[1] Menon, S., & Vondrick, C. (2023). Visual Classification via Description from Large Language Models. In ICLR 2023.
[2] Novack, Z., McAuley, J., Lipton, Z. C., & Garg, S. (2023, July). Chils: Zero-shot image classification with hierarchical label sets. In ICML 2023.
[3] Yang, Y., Panagopoulou, A., Zhou, S., Jin, D., Callison-Burch, C., & Yatskar, M. (2023). Language in a bottle: Language model guided concept bottlenecks for interpretable image classification. In CVPR 2023.
[4] Roth, K., Kim, J. M., Koepke, A. S., Vinyals, O., Schmid, C., & Akata, Z. (2023, October). Waffling Around for Performance: Visual Classification with Random Words and Broad Concepts. In ICCV 2023
[5] Udandarao, V., Gupta, A., & Albanie, S. (2023). Sus-x: Training-free name-only transfer of vision-language models. In ICCV 2023

---

> ### Author Response · Authors · 2023-11-20
> **To all reviewers (2/2)**
>
> ## Other VLMs
>
> Multiple reviewers raised concerns about our choice of CLIP as our VLM. We emphasize that our main focus is on generating informative natural language descriptions that improve image classification.
>
> To address reviewers’ concerns, we show that the generated descriptions provide useful information that is compatible with a wide range of VLMs. Here, we provide additional results on two OpenClip models[2], which are significantly stronger than their CLIP counterparts (e.g., 4 percentage points improvement on ImageNet with a single template compared to CLIP for ViT-L/14), as well as BLIP2[1], which uses an entirely different approach for training VLMs.
>
> As shown below, the generated descriptions are useful for these models as well and further improve the performance of naive descriptions. Please see the updated manuscript for experiments on other datasets.
>
>
> |Dataset|EuroSAT|EuroSAT|EuroSAT|EuroSAT|Flowers|Flowers|Flowers|Flowers|
> |---|---|---|---|---|---|---|---|---|
> |Model|ViT-B-32|ViT-B-32|ViT-L-14|BLIP2|ViT-B-32|ViT-B-32|ViT-L-14|BLIP2|
> |Pretraining|DComp|Laion|DComp|-|DComp|Laion|DComp|-|
> |Single Template|38.03|41.73|61.14|52.14|72.13|67.49|80.60|55.91|
> |Template Set|41.02|41.44|61.62|51.01|72.06|67.67|81.12|57.85|
> |Naive|49.28|45.37|69.72|63.63|73.18|66.53|79.90|60.81|
> |FuDD (K=10)|55.74|57.27|74.05|70.84|73.98|69.00|83.35|61.60|
> |FuDD (K=C)|55.74|57.27|74.05|70.84|75.05|69.91|83.02|61.21|
>
>
> DComp is the checkpoint trained on the datacomp_xl_s13b_b90k dataset, and Laion is the checkpoint trained on laion2b_e16 [3].
>
> [1] Li, J., Li, D., Savarese, S., & Hoi, S. (2023). BLIP-2: Bootstrapping Language-Image Pre-training with Frozen Image Encoders and Large Language Models. ICML.
> [2] Cherti, M., Beaumont, R., Wightman, R., Wortsman, M., Ilharco, G., Gordon, C., … Jitsev, J. (2023). Reproducible scaling laws for contrastive language-image learning. Proceedings of the IEEE/CVF Conference on Computer Vision and Pattern Recognition, 2818–2829.
> [3] https://github.com/mlfoundations/open_clip
>
>
> ## API Costs
>
> Although FuDD queries the LLM more than naive approaches, the API calls are pretty affordable and do not hinder the wider adoption of FuDD. On average, one input prompt and model response are 380 and 199 tokens, respectively. With OpenAI pricing at the time of writing (0.001 and 0.002 dollars per 1k tokens for input and response),  the cost is \$0.78 per 1000 queries, leading to affordable prices as mentioned below (prices are in dollars).
>
> |Cub|DTD|EuroSAT|FGVCAircraft|Flowers|Food101|ImageNet|Pets|Places365|Stanford Cars|Stanford Dogs|
> |---|---|---|---|---|---|---|---|---|---|---|
> |15.47|0.84|0.03|3.85|4|3.93|60.24|0.52|51.63|14.85|5.55|
>
> FuDD also accommodates datasets with a large number of classes like ImageNet by recognizing the more significant role of ambiguous classes, reducing the costs for the ImageNet dataset from 388 to 60 dollars (See appendix B for details).
>
> In addition, as studied extensively in our work, we can use off-the-shelf or fine-tuned LLMs like Llama 2 to generate differential descriptions using in-house hardware to avoid API costs or accommodate other issues like working with private and sensitive data.

---

### Meta-Review · Area_Chair_iEtp · 2023-12-05

**Metareview:**

Sentiment among reviewers is overall mildly positive, with numerous concerns/questions, including about novelty. However, the idea is fairly interesting, with reasonable (albeit small) gains.

**Justification For Why Not Higher Score:**

The only 8 is from a reviewer who seems not too familiar with the field

**Justification For Why Not Lower Score:**

Worst score is 5, with an 8 and 6

---

### Decision · Program_Chairs · 2024-01-16

Accept (poster)